# Efficacy of Loop-Mediated Isothermal Amplification for *H. pylori* Detection as Point-of-Care Testing by Noninvasive Sampling

**DOI:** 10.3390/diagnostics11091538

**Published:** 2021-08-25

**Authors:** Amir Sohrabi, Joar Franzen, Nikolaos Tertipis, Ulrika Zagai, Wanxin Li, Zongli Zheng, Weimin Ye

**Affiliations:** 1Department of Medical Epidemiology and Biostatistics, Karolinska Institutet, Stockholm 17165, Sweden; amir.sohrabi@ki.se (A.S.); joar.franzen@ki.se (J.F.); nikolaos.c.tertipis@gmail.com (N.T.); ulrika.zagai@ki.se (U.Z.); 2School of Public Health, Fujian Medical University, Fujian 350122, China; liwx2012@126.com; 3Department of Biomedical Sciences, City University of Hong Kong, Hong Kong, China; zongli.zheng@ki.se

**Keywords:** *Helicobacter pylori*, gastric cancer, noninvasive sampling, LAMP, POCT

## Abstract

For targeted eradication of *Helicobacter*
*pylori* (*H. pylori*) to reduce gastric cancer burden, a convenient approach is definitely needed. The purpose of this study was to evaluate the LAMP assay for *H. pylori* detection using samples collected by noninvasive and self-sampling methods. The available LAMP assay for *H. pylori* detection was appraised and verified using reference and clinically isolated *H. pylori* strains. In addition, a clinical study was conducted to assess the LAMP assay on 51 patients, from whom saliva, oral brushing samples, feces, corpus, and antrum specimens were available. Clarithromycin resistance was also analysed through detection of A2143G mutation using the LAMP-RFLP method. The validation and verification analysis demonstrated that the LAMP assay had an acceptable result in terms of specificity, sensitivity, reproducibility, and accuracy for clinical settings. The LAMP assay showed a detection limit for *H. pylori* down to 0.25 fg/µL of genomic DNA. An acceptable consensus was observed using saliva samples (sensitivity 58.1%, specificity 84.2%, PPV 85.7%, NPV 55.2%, accuracy 68%) in comparison to biopsy sampling as the gold standard. The performance testing of different combinations of noninvasive sampling methods demonstrated that a combination of saliva and oral brushing could achieve a sensitivity of 74.2% and a specificity of 57.9%. A2143G mutation detection by LAMP-RFLP showed perfect consensus with Sanger sequencing results. It appears that the LAMP assay in combination with noninvasive and self-sampling as a point-of-care testing (POCT) approach has potential usefulness to detect *H.*
*pylori* infection in clinic settings and screening programs.

## 1. Introduction

*Helicobacter**pylori* (*H. pylori*) infection is one of the most common infections, infecting over 50% of the world’s population. The current hypothesis is that *H.*
*pylori* has co-evolved with humans and acts as an important part of the normal gastric flora. *H. pylori* is a gram-negative, spiral-shaped, microaerophilic bacterium able to cause diseases from chronic gastritis and gastric and duodenal ulcers to mucosa-associated lymphoid tissue (MALT) lymphoma and gastric adenocarcinoma [1]. The International Agency for Research on Cancer (IARC) has classified *H. pylori* as a first-class carcinogen. Based on few clinical trials and population-based observational studies, widespread eradication has been proposed for primary prevention of stomach cancer. However, this approach may be counterproductive. First, widespread use of antibiotics would induce problems with multidrug resistance. Second, the accumulated epidemiologic evidence for an inverse relation between *H. pylori* infection and the risk of esophageal adenocarcinoma emphasizes the necessity to avoid unnecessary eradication. Some infected subjects, notably those with duodenal ulcers, appear to have a much lower risk of stomach cancer than the population average. Thus, risk stratification of the population for targeted primary prevention is critical, and development of an easy-to-use method for *H. pylori* subtyping, especially in low-resources settings, would be an important step. Currently, *H. pylori* detection methods are divided into invasive and noninvasive procedures. Invasive methods rely on endoscopic examination and biopsies, which has notable drawbacks, such as invasive and inconsistent sampling, resource intensiveness, low cost effectiveness, and inconvenience, especially for elderly or severely ill subjects. Traditionally, the presence of *H. pylori* is tested by rapid urease test. However, many other bacteria are capable of producing urea. Subtyping of *H. pylori* usually relies on culture-based procedures, which usually have a low sensitivity and are time-consuming and challenging. Molecular diagnostic technologies are always in the forefront of pathogen detection, although the availability as well as the expensiveness of the facilities and instruments can be considered as disadvantages of these technologies [2,3,4,5,6,7,8]. 

Developing a rapid, noninvasive, and cost-effective approach for the genotyping of *H. pylori* using oral brushing or fecal samples, instead of using endoscopic biopsy samples, will still be critical for implementation of the targeted eradication. Furthermore, to choose effective antibiotics for eradication, genotypic profiling of antibiotic resistance among *H. pylori* mutants will also be important. Developing noninvasive tools for the identification of infections with highly virulent *H. pylori* strains or other microorganisms are needed to serve the targeted chemoprevention and eradication program. Ideally, extending point-of-care testing (POCT) and rapid DNA extraction, plus a thermocycler-free *H. pylori* genotyping method using either oral or fecal samples, would suit the prevention program best. A loop-mediated isothermal amplification (LAMP) for specific amplification of nucleic acids might serve this purpose, which is characterised by its high specificity due to the use of six primers, high sensitivity, around 10-fold shorter reaction time, isothermal gene amplification without the need of a thermocycler or high-throughput facilities, and the possibility of detection with the naked eye [3,6,7,8,9,10,11].

We have evaluated and verified LAMP efficacy as point-of-care testing to detect *H. pylori* using noninvasive sampling with the collection of saliva, oral brush, and fecal samples instead of an invasive approach, e.g., the collection of gastric corpus and antrum specimens. In addition, an endoscopy room-based study was conducted to validate the usefulness of the developed protocol. We also assessed A2143G mutation detection for analysing clarithromycin resistance as the first choice of *H. pylori* treatment.

## 2. Materials and Methods

### 2.1. H. pylori Strains and Clinical Samples

*H. pylori* reference strains and 42 clinical *H. pylori* strains isolated from gastric biopsy specimens were used for evaluation of performance and efficacy of the LAMP approach and LAMP-RFLP assay. The reference strains of *H. pylori* include HPAG1, DU30, 26695, PMSS1, J99, G27, PMSST2, and CCUG30. DNA of these reference strains was extracted using the QIAmp DNA mini kit (Qiagen©, Hilden, Germany). A DNA sample was also constructed to represent the mutant 26,695 strain for testing the LAMP-RFLP assay by Eurofins genomics© (Ebersberg, Germany). 

Among 42 clinical isolates, 27 were verified by colony characteristics and a urease test (DNA was extracted using the bacterial genomic DNA extraction kit, TIANGEN©, Beijing, China) at Fujian Medical University, China, and 15 were identified by colony characteristics and staining (DNA was extracted using the bacterial genomic DNeasy Mini extraction kit, Qiagen©) at Karolinska Institutet, Sweden. All of the 42 *H. pylori* clinical strains and *H. pylori* reference strains, HPAG1, DU30, and 26,695, were also analyzed by Sanger sequencing (23S rRNA) (Table 1) [12,13,14,15].

### 2.2. Validation and Verification of the LAMP Method

The *ure*C gene of *H. pylori* was used as a target gene for LAMP reactions [11,12]. Sequences from multiple strains of *H. pylori*, including HPAG1, 26695, J99, G27, DU30, and PMSS1, were retrieved and verified by the GenBank database on NCBI. We used the primer sequences developed by Bakhtiari et al. with the primer explorer version 5 (www.primerexplorer.jp, accessed on 2 October 2017) [7,8]. The sequences of the primers are stated in Table 1. The total volume of the LAMP reaction was 30 µL. The mixture consisted of 1.6 µM of FIP/BIP, 0.2 µM of F3/B3, 0.4 µM of LF/LB, 1.4 mM of dNTPs, 10X reaction buffer (NEB), 1 µL of Bst 2.0 DNA polymerase, and 1 µL of DNA template. For visual detection of the reaction results, 1 µL of 1/10 diluted original SYBR Green I 10000X (Thermo Fisher Scientific©, Waltham, MA, USA) was added to the inner lid of the reaction tube prior to the reaction. Upon amplification, tubes were inverted several (5–10) times or vortexed and quick spun for mixing the dye with the amplified products. The colour change allowed naked-eye detection. 

For achieving the ideal outcome of the LAMP reaction, we optimized the temperature, reaction time, and concentration of mixture components. We tested the combinations of different temperatures (60 °C up to 65 °C) and amplification times (30 to 60 min). The optimal temperature and time were determined to be 65 °C for 60 min, and this was selected for amplification of the LAMP procedure.

We further checked whether the LAMP assay could detect *H. pylori* without DNA extraction. Two colonies of HPAG1 and DU-30 strains were dissolved in 20 µL of d.d.H_2_O, followed by incubation at 95 °C for 7 min, followed by the LAMP assay.

In order to indicate the presence of transversion mutation A > G in 23S rRNA of *H. pylori*, we further assessed the LAMP restriction fragment length polymorphism (LAMP-RFLP) assay. We used the restricting enzymes Ava II (Eco47I) and BsaI (Eco31I) (New England Biolabs©, Ipswich, MA, USA). Briefly, Ava II and BsaI digest all amplification products of LAMP using specific primers for A2143G mutation detection (Table 1). To check the performance of the LAMP-RFLP assay, we used *H. pylori* reference strains and 42 *H. pylori* strains isolated from gastric biopsy samples. Digestion patterns can be distinguished as wild type or mutation profile by 3% agarose gel electrophoresis [16].

### 2.3. Evaluation of Noninvasive Sampling for Detection of H. pylori Using LAMP

Detection of *H. pylori* is convenient by collecting oral brushing, saliva, or fecal samples via noninvasive sampling compared to the invasive biopsy in the stomach. However, a biopsy is commonly used as the gold standard of sampling for *H. pylori* identification. We considered the presence of *H. pylori* in the antrum or corpus as the gold standard, and compared results from oral brushing, saliva, and fecal specimens with this gold standard.

Thirty-five women and sixteen men were recruited from an endoscopy clinic at the Ersta Hospital in Stockholm, Sweden, between February 2018 and January 2021. All 51 subjects were between 20 and 64 years old (average age was 44.81 ± 13.29), who were referred for endoscopy at the hospital no matter the diagnosis. The exclusion criteria for patients were, among others, earlier surgical removal of parts of the stomach or esophagus, narrowing or blockage from the esophagus or stomach, a known or suspected bleeding disorder, reduced general condition, and/or complicated diseases that can risk even a small extension of examination time and pregnancy. Each patient was informed about the objectives of the study and signed a consent form before participating in the study. This study was approved by the Stockholm Ethics Vetting Board (DNR. No. 2017/1453-31). Saliva samples were collected after at least 30 min of no eating, drinking, smoking, brushing teeth, or chewing gum. The patients were asked to rub their cheeks carefully for 30 s, and then spit 2–3 mL saliva into a sterile 50 mL tube; 1.3 mL was transferred to a 1.5 mL micro tube with a sterile pipet. Oral brushes were collected with a disposable sterile cytology brush. The left side of the oral cavity was brushed 10 times over an area of 2 × 2 cm. The handle of the brush was cut off with sterile pliers, and the head of the brush was placed in a 1.5 mL micro tube. In addition, biopsy samples (antrum and corpus) were taken from the greater curvature. All of the samples were immediately kept in −80 °C until experiments. Furthermore, fecal samples were collected by themselves at home using a DNA/RNA Shield™ Fecal Collection Tube. These fecal samples were mailed to the study center at Karolinska Institutet, Stockholm, Sweden, and kept frozen in −80 °C until DNA extraction. In total, 51 saliva samples, 51 oral brushing samples, 50 biopsy samples from the antrum and corpus each, and 43 fecal specimens were collected for the present study. DNA from clinical samples (antrum/corpus biopsy samples, oral brushing, and saliva specimens) was extracted using a QIAmp DNA mini kit (Qiagen©, Hilden, Germany). The fecal specimens were extracted by a PowerFecal Pro DNA Kit (Qiagen©, Hilden, Germany). We used SAS 9.4 to calculate sensitivity, specificity, positive predictive value (PPV), negative predictive value (NPC), and accuracy.

## 3. Results

### 3.1. Analysis of the Performance of LAMP and LAMP-RFLP

The specificity of the LAMP reaction for detecting *H. pylori* was assessed using genomic DNA isolated from different *H. pylori* reference strains HPAG1, 26,695, PMSS1, and DU 30, as well as 42 clinical strains. As negative controls, other bacteria were used, such as *E. coli*, *Lactobacilli*, and *Salmonella*. The specificity of the designed primers was evaluated by both LAMP and conventional PCR (*UreC*/*GlmM)* methods. The detailed data of conventional PCR are not shown. Each reaction that turned green upon amplification was considered positive and was confirmed with gel electrophoresis, where the ladder-like pattern was indicative of positivity (Figure 1 and Appendix A). The analysis of LAMP specificity showed there were no non-specific amplification bands and false positive results in any gel electrophoresis when testing the *H. pylori* reference strains and 42 clinical *H. pylori* strains isolated from gastric biopsy specimens.

The analytical sensitivity and detection limit of the LAMP assay were assessed by a series of 10-fold dilutions of the *H**. pylori* HPAG1 strain in comparison to normal PCR. Qubit^®^ 2.0 Fluorometer (Invitrogen©/Thermo Fisher Scientific©, Waltham, MA, USA) was used for the measurement of extracted DNA concentration. The samples were measured three times, and the mean value was considered as the real concentration of the specimens. Upon measurement, serial dilutions were created from 0.7 ng/µL to 0.125 fg/µL. For these reactions, 1 µL of each sample was used as a template (Figure 2). The LAMP assay showed a detection limit for *H. pylori* down to 0.25 fg/µL of genomic DNA, while conventional PCR could only detect *H. pylori* down to 0.01 ng/µL, a 40,000× fold difference. The reproducibility test revealed that the results were similar in triplicate or duplicate runs. The data were consistent between extracted DNA and the extraction-free procedure of HPAG1 and DU30 strains as templates (Figure 1, Figure 2, and Appendix A).

The LAMP-RFLP assay could clearly distinguish wild-type *H. pylori* strains and the synthetic mutant 26,695 strain (Figure 3A). Sanger sequencing consequences demonstrated that 9 out of 27 Chinese strains (33.3%) showed an A2143G mutation in the 2143 region, whereas all 15 *H. pylori* clinical strains from Sweden were sensitive to clarithromycin. Sanger sequencing results from an A2143G mutant sequence and HP 26,695 (wild type) as references, accompanied with 12 strains from China, are shown in Figure 3B. After running the LAMP-RFLP assay, the LAMP-RFLP results were 100% consistent with the Sanger sequencing outputs (Figure 3).

### 3.2. Results of an Endoscopy-Based Study Comparing Different Sampling Methods to Detect H. pylori by LAMP

The LAMP assay was performed on 245 clinical samples from 51 participants in Ersta Hospital of Sweden. From these, 115 (47%) samples were positive for *H. pylori*. Analyses by sample type revealed that *H. pylori* could be detected by LAMP among 41.2% of saliva, 53% of oral brush, 56% of corpus, 48% of antrum, and 34.8% of fecal samples (Appendix A). The analysis of A2143G mutation using LAMP-RFLP showed that all of them were susceptible to clarithromycin.

The result comparison showed that the noninvasive sampling had acceptable analytical performance compared to invasive biopsy sampling (antrum or corpus as the gold standard). The best results were observed using saliva samples (sensitivity 58.1%, specificity 84.2%, PPV 85.7%, NPV 55.2%, accuracy 68%), followed by oral brushing samples and fecal samples (Table 2). We also tested the performance of different combinations of noninvasive sampling methods. The highest sensitivity was achieved by treating the three sampling methods as positive (sensitivity 80.7%, specificity 47.4%), followed by the combination of saliva and oral brushing (sensitivity 74.2%, specificity 57.9%), although the combination of fecal samples with oral samples did not significantly improve performance (data not shown).

## 4. Discussion

*H. pylori* is a common infectious agent that has been acknowledged as the most important risk factor of gastric cancer. *H. pylori* testing is important for stomach cancer screening programs [2,17]. There are different diagnostic tools for *H. pylori* detection, including both invasive and noninvasive methods. Endoscopic examination with a urease test on tissue specimens is categorized as an invasive method. On the other hand, the urea breath test (UBT), stool antigen test (SAT), and antibody and molecular detection on saliva, stool, and urine are classified as noninvasive procedures [5,17,18,19,20]. In the past few decades, molecular point-of-care testing for infectious pathogen detection has been developing to be used in field investigation, at the patient’s bedside, and in the physician’s office. The molecular POCTs as gene amplification methods have been increasingly used in clinical diagnostics and health surveillance systems for population screening. Implementation of a new generation of POCTs can have critical effects on reducing the time in amplification of small target genes and better decision-making for treatment [21,22,23,24].

The present study results suggested that the LAMP method for detection of *H. pylori*, the most important pathogen involved in gastric cancer, could be considered as a rapid diagnostic test. Amplification of *Ure*C gene of *H. pylori* during 60 min is an interesting point for clinicians. Utilization of a single enzyme (Bst DNA polymerase), iso-thermal condition, and interpretation by the naked eye without any complex instruments, as well as amplification of the *H. pylori* genome without any extraction procedures, are advantages of this POCT. The ability of detection of 0.25 fg/µL of the *H. pylori* DNA genome from clinical samples shows its high sensitivity for pathogen diagnosis.

Six primers are used in the LAMP procedure for amplification of the target gene. This avoids non-specific amplification bands. It seems that the specificity and accuracy of LAMP are higher than conventional PCR and other immunological rapid tests. These characteristics of the LAMP assay were in accordance with other studies. However, some difficulties were encountered; for instance, the usage of six primers and their interaction with each other lead to the production of small dsDNA. The false-positive results increased due to these interactions of primers. As a result, our validation and verification results showed that the LAMP assay might be a useful diagnostic tool. However, it is also suggested to compare the LAMP assay results with approved *CE* marked and in vitro diagnostic (IVD) kits to verify the outcomes in the future studies. 

When we applied the LAMP assay to samples collected in an endoscopy-room-based study, the results showed that *H. pylori* can be easily detected from noninvasively collected samples, such as saliva, oral brushing and fecal samples. The results were comparable with those detected using invasively collected samples, like gastric corpus and antrum tissues. The analytical performance specifications of noninvasive sampling demonstrated that saliva might be the most suitable, and adding oral brushing might further improve the performance. It would appear the invasive biopsy sampling could be replaced by noninvasive and oral self-sampling, albeit it needs to be further tested by comprehensive studies with larger sample size. Moreover, applying a molecular POCT for detection of microbes and any drug resistance are helpful in selecting the best treatment strategy in low-resource countries.

Future studies should aim at improving the suitability of SNPs detection and other variants of *H. pylori* genome, which might facilitate risk-stratification for targeted eradication and selection of appropriate antibiotics. The current LAMP-RFLP method is difficult to be applied in clinical settings or population-based screening and eradication programs.

To summarize, our results suggested that the LAMP method has acceptable performance, reliability, and suitability for *H*. *pylori* detection in the management of gastrointestinal disorders and gastric cancer screening. Amplification of the target gene in 1 h without any need for genome extraction or specific instruments as a user-friendly, cost-effective method, and subsequent simple measuring results with the naked eye, makes the LAMP assay for *H. pylori* detection especially suitable in low-income countries. The noninvasive and pain-free self-sampling of saliva and oral brushing further increase its applicability. It would appear that LAMP could be applied as point-of-care testing for *H. pylori* detection, and, in combination with noninvasive self-sampling, it can be applied widely in clinical settings and gastric cancer screening programs. 

## Figures and Tables

**Figure 1 diagnostics-11-01538-f001:**
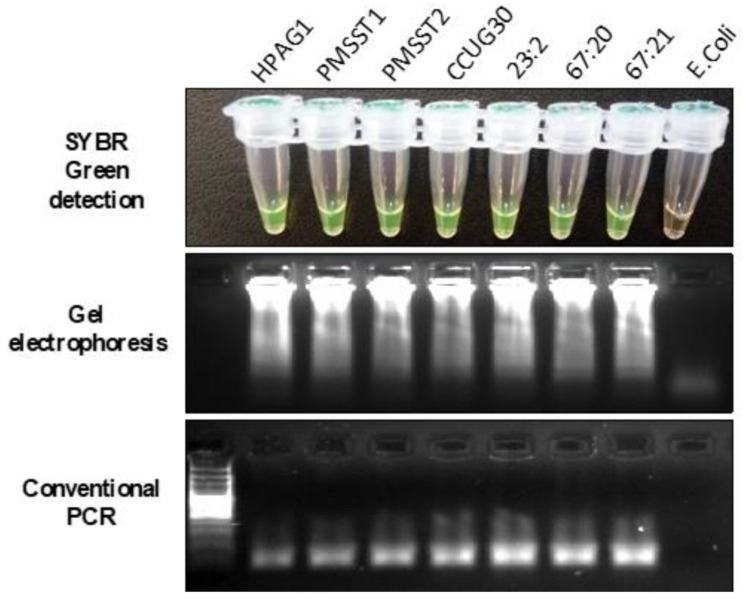
Lanes HPAG1, PMSST1, PMSST2, and CCUG30 (reference strains), 23:2, 67:20, 67:21 (patients) show different *H. pylori* strains. Lane *E. coli* was used as a negative control for testing the specificity of the assay. All of the lanes except *E. coli* show amplification by naked eye, by gel electrophoresis of the LAMP products, and by conventional PCR.

**Figure 2 diagnostics-11-01538-f002:**
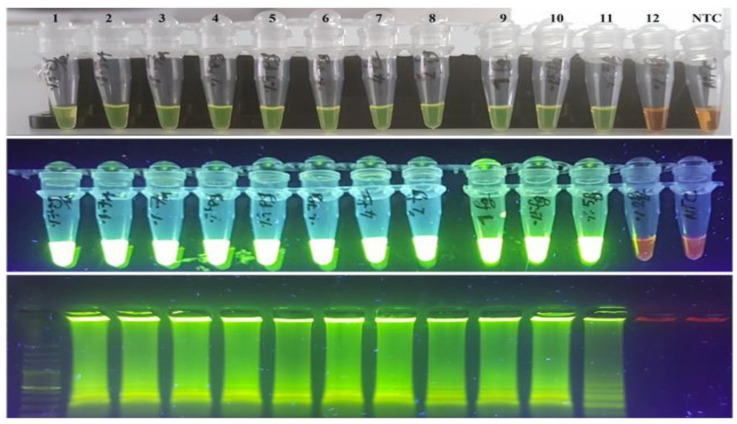
Detection limits of the LAMP assay are shown by SYBR Green I, naked eye, under UV and 3% gel electrophoresis (Ladder 100bp), respectively. Lane 1: HPAG1 DNA (0.7 ng/µL), Lane 2: 0.07 ng/µL, Lane 3: 0.007 ng/µL, Lane 4: 0.7 pg/µL, Lane 5: 0.07 pg/µL, Lane 6: 0.007 pg/µL, Lane 7: 4 fg/µL, Lane 8: 2 fg/µL, Lane 9: 1 fg/µL, Lane 10: 0.5 fg/µL, Lane 11: 0.25 fg/µL, Lane 12: 0.125 fg/µL and None-Template Control (NTC).

**Figure 3 diagnostics-11-01538-f003:**
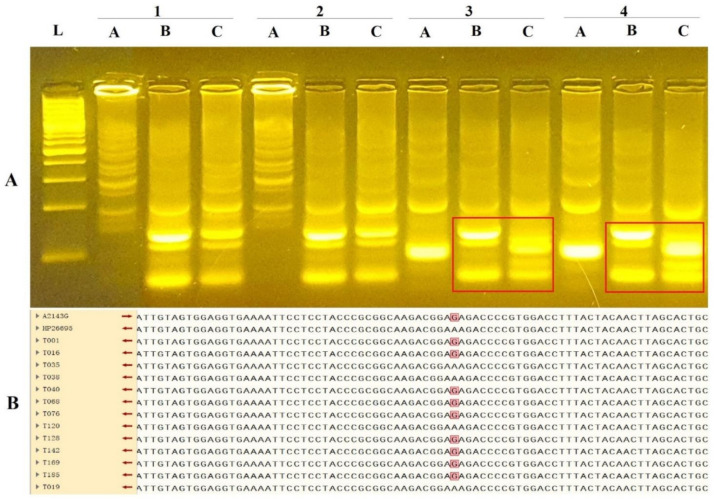
LAMP-RFLP and Sanger sequencing results for the detection of clarithromycin resistance in the validation and verification process. The A, B, and C arrows mean BsaI, AvaII, and BsaI and AvaII digestions, respectively. The pattern of mutant variants differs from wild types in the B and C lines for each subject on the 3% agarose gel electrophoresis. (**A**) LAMP-RFLP verification process. L: Ladder 100 bp; 1-DU30 strain (wild type), 2-HPAG1 strain (wild type), and 3,4-26,695 strain (synthetic mutant strain) (A > G). (**B**) Sequence alignment showing A2143G mutation in the 2143 region based on Sanger sequencing results (an A2143G mutant sequence and HP 26,695 (wild type) as references). The results of Chinese isolates are listed, including nine with an A2143G mutation. The red box demonstrates position of point mutation (A > G).

**Table 1 diagnostics-11-01538-t001:** Primer sequences of the *H. pylori ureC* gene for LAMP amplification, *GlmM* for conventional PCR, and LAMP-RFLP for A2143G mutation.

Primers Name	Sequence (5′ > 3′)
ureC-FIP	GCA TAT CAT TTT TAG CGA TTA CGC TCA CTA ACG CGC TCACTT G
ureC-BIP	CTC GCC TCC AAA ATT GGC TTG CGA TTG GGG ATA AGTTTG
ureC-F3	GCT TAC CTG CTT GCT TTC
ureC-B3	TCC CAA GAT TTG GAA TTG AAG
ureC-LF	CAG GCG ATG GTT TGG TGT G
ureC-LB	TCA ATT GCA TGC ATT CGC TCA
GlmM-F	GCT CACTAAAG CGTTTTC TACCATAT AG
GlmM-R	ATTGCTGCCGGATTGTATTTTAA
23r-F	TCAAACTACCCACCAAGCATTGTCC
23r-R	CGAAGGTTAAGAGAATGCGTCAGTC
F3	ACCGACCTG CATGAATGG
B3	AGCCAAAGCCCTTACTTCAA
LF	CCTCCACTACAATTTCACTGAATCT
LB	ACAACTTAGCACTGCTAATGGGAAT
FIP	GCCGCGGGTAGGAGGAATTTTCGTAACGAGATGGGAGCTGTC
BIP/Wild type	CGGAAAGACCCCGTGGACCTAGCCTCCCACCTATCCTG
BIP/Mutant	ACCCCGTGGACCTTTACAGCCTCCCACCTATCCTG

**Table 2 diagnostics-11-01538-t002:** Comparison of *H. pylori* (HP) diagnosis outcomes between gold standard sampling (biopsy from antrum/corpus) and noninvasive sampling, assayed by LAMP.

HP Result by Noninvasive Sampling	*H. pylori* Detected by Gold Standard Sampling (Antrum/Corpus)			Analytical Performance (95% CI)
	HP Positive	HP Negative	Total	
Saliva (HP Positive)	18	3	21	^3^ PPV: 85.7 (70.8–100)
Saliva (HP Negative)	13	16	29	^4^ NPV: 55.2 (37.1–73.3)
Total	31	19	50	
Analytical Performance (95% CI)	^1^ Sen: 58.1 (41.7–75.4)	^2^ Spe: 84.2 (67.8–100)	Accuracy: 68%	
Oral Brushing (HP Positive)	19	8	27	PPV: 70.4 (53.2–87.6)
Oral Brushing (HP Negative)	12	11	23	NPV: 47.8 (27.4–68.2)
Total	31	19	50	
Analytical Performance (95% CI)	Sen: 61.3 (44.1–78.4)	Spe: 57.9 (35.7–80.1)	Accuracy: 60%	
Fecal (HP Positive)	9	5	14	PPV: 64.3 (39.2–89.4)
Fecal (HP Negative)	18	10	28	NPV: 35.7 (18.0–53.5)
Total	27	15	42	
Analytical Performance (95% CI)	Sen: 33.3 (15.6–51.1)	Spe: 66.7 (42.8–90.5)	Accuracy: 45.3%	

^1^ Sen: sensitivity (%). ^2^ Spe: specificity (%). ^3^ PPV: positive predictive value (%). ^4^ NPV: negative predictive value (%).

## Data Availability

All data are mentioned in the body of the manuscript, tables, figures and supplementary files.

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
