# Peer review of "Efficacy of Loop-Mediated Isothermal Amplification for H. pylori Detection as Point-of-Care Testing by Noninvasive Sampling"

_diagnostics, 2021, doi:10.3390/diagnostics11091538_

Round 1

Reviewer 1 Report

The manuscript of Sohrabi et al entitled “Efficacy of loop-mediated isothermal amplification for H. pylori detection as point-of-care testing by non-invasive sampling” reported the evaluation of a loop-mediated isothermal amplification (LAMP) assay for H. pylori detection.

The LAMP assay was validated in vitro using reference and clinically isolated strains of H. pylori and some non-H. pylori strains. In addition, a clinical study that included 51 patients who provided saliva, oral brushing, feces, corpus and antrum specimens were used to assess the presence of H. pylori using LAMP. Also the authors assessed for the presence of the A2143G mutation using LAMP-RFLP. The authors found that sensitivity and specificity of LAMP in vitro was excellent. However, their results of sensitivity and specificity in the clinical setting was promising but not great. The sensitivity of LAMP was less than 62%  with the oral brushing samples but the specificity was less than 58%. The use of saliva had a better specificity (84.2%) but worse sensitivity (58.1%). Results of the feces samples were even worse than using the oral cavity samples. The authors used as gold standard the LAMP results in the antrum and corpus biopsies. The authors concluded that LAMP assay has a potential usefulness to detect H. pylori infection in a clinical setting.

This is an interesting study that reported the use of LAMP assay as an affordable and easy to use method on the basis that it is thermocycler free and the result can be detected with the naked eye. The results of the assessment of the LAMP assay were excellent for the in vitro studies. However, the results of the clinical setting were less impressive. The sensitivity and specificity is promising but not great particularly in countries where urea breath test is available. In addition there are some points that require some clarification.

  1. Why the author included in materials and methods the results of the validation and verification of LAMP?
  2. On figure 3, the third panel should be eliminated and only mentioned in the results section. The authors must include a better gel for figure 3 section A.
  3. The authors mentioned the use of H. pylori colonies without DNA extraction in the LAMP assay, however, the results were not included neither the detection limits of the assay.
  4. In the first paragraph of the results section the authors said that non non-specific amplification bands were observed in all gel electrophoresis when testing H. pylori reference strains and the 42 clinical H. pylori strains. However the presence or not of none-specific amplification should be observed and reported in non-H. pylori DNA. Please clarify.
  5. The major problem with the study is that the authors used the same assay (LAMP) in antrum and corpus biopsies as the gold standard. This approach is incorrect and the authors must use a different well validated methodology for the detection of H. pylori in gastric biopsies as gold standard. Furthermore, according with the supplemental table provided by the authors only 20 patients were positive in both biopsies with LAMP (true positive) and 19 were negative for both biopsy samples (true negative). The authors need to report based on those numbers and the possible positive (n=11) should be discuss apart.

Overall the LAMP assay is very promising  but the assessment of sensitivity and specificity in the clinical setting in this study is unclear and the authors need to make the appropriate modifications

Author Response

Dear Reviewers,

Initially, we would like to thank reviewers for their comments, which will have been improved the manuscript significantly. We have revised the manuscript based on the comments and corrections are included and explained as follow. All reviewer’s comments and corrections are also shown in yellow highlight style throughout the manuscript. We also add some corrections in the manuscript for better scientific writing style, English grammatical and typos as well.

Yours Sincerely,

Reviewer 1

a. Why the author included in materials and methods the results of the validation and verification of LAMP?

- The results of validation and verification of LAMP method are moved to result section.

b. On figure 3, the third panel should be eliminated and only mentioned in the results section. The authors must include a better gel for figure 3 section A.

-  Quality of figure 3 is improved and corrected. Section B2 is eliminated and only mentioned in the manuscript.

c. The authors mentioned the use of H. pylori colonies without DNA extraction in the LAMP assay, however, the results were not included neither the detection limits of the assay.

- We have only tested the H.pylori HPAG1 and DU-30 strains by LAMP assay without extraction. It is mentioned in line 110-112. We have not tested on clinical samples and other reference strains for some limitations like insufficient samples and isolates.

d. In the first paragraph of the results section, the authors said that non non-specific amplification bands were observed in all gel electrophoresis when testing H. pylori reference strains and the 42 clinical H. pylori strains. However the presence or not of none-specific amplification should be observed and reported in non-H. pylori DNA. Please clarify.

- We meant there were not any non-specific bands in samples that were positive for H.pylori. In addition, there were not any cross-reactions and false positive results by optimization of the reaction mixture and amplification condition. So, the non-specific bands were so faint and invisible.

e. The major problem with the study is that the authors used the same assay (LAMP) in antrum and corpus biopsies as the gold standard. This approach is incorrect and the authors must use a different well-validated methodology for the detection of H. pylori in gastric biopsies as gold standard. Furthermore, according with the supplemental table provided by the authors only 20 patients were positive in both biopsies with LAMP (true positive) and 19 were negative for both biopsy samples (true negative). The authors need to report based on those numbers and the possible positive (n=11) should be discuss apart.

 - Firstly, we validated and verified the LAMP method comprehensively as described throughout the manuscript and compared the results by conventional PCR and Sanger sequencing as well by H.pylori reference strains and approved clinical isolates. In addition, some corpus and antrum are sequenced by Sanger sequencing meanwhile the validation step and results were consistent to LAMP assay.

The corpus and antrum samplings were considered as gold standard sampling for H.pylori detection. The LAMP results were compared between different kinds of samples as well. As corpus and antrum regions are next to each other, so in endoscopic examination and biopsies are so complicated to collect tissues properly in proper condition and it might be taken incorrectly. In addition, the dimension of corpus and antrum samples were so small and it can be effect on result by insufficient samples. Therefore, we have decided to consider positivity of the corpus or antrum for H.pylori, however, if corpus + or antrum – and corpus – and antrum +.

Reviewer 2 Report

The authors attempted to design and test the LAMP method for the detection of H. pylori. The design of the experiment and its description found sound. However, I still find few small concerns in the study, which are enumerated below:

  1. Please define POCT, when you are using it for the first time.
  2. In Line 85-86: The authors state that Eurofins constructed a synthetic mutant strain. Would you please correct the sentence to a DNA constructed to represent the mutant for testing the LAMP assay?
  3. In Line 113-114: The authors used few bacteria as negative controls. I am unsure whether these bacteria have urease or are urease negative. If these bacteria are urease negative, could authors test their assay with the DNA of non-Helicobacter bacteria expressing urease as a negative control?
  4. In figure 2, authors showed their assay could detect 0.125 fg of DNA. Could authors dilute the template further to get to the point they could not see the bacterial DNA to know what would be the least amount of DNA required for its detection? Additionally, when diluting the target DNA could they also spike the samples with non-specific DNA to ensure The non-specific DNA does not mask H pylori detection.

Author Response

Dear Reviewers,

Initially, we would like to thank reviewers for their comments, which will have been improved the manuscript significantly. We have revised the manuscript based on the comments and corrections are included and explained as follow. All reviewer’s comments and corrections are also shown in yellow highlight style throughout the manuscript. We also add some corrections in the manuscript for better scientific writing style, English grammatical and typos as well.

Yours Sincerely,

Reviewer 2

1. Please define POCT, when you are using it for the first time.

- It is corrected in the abstract where it was using for the first time.

2. In Line 85-86: The authors state that Eurofins constructed a synthetic mutant strain. Would you please correct the sentence to a DNA constructed to represent the mutant for testing the LAMP assay?

- It is corrected in the line 85-86 of manuscript.

3. In Line 113-114: The authors used few bacteria as negative controls. I am unsure whether these bacteria have urease or are urease negative. If these bacteria are urease negative, could authors test their assay with the DNA of non-Helicobacter bacteria expressing urease as a negative control?

- We have tested some bacteria for confirmation the specificity of LAMP method. Unfortunately, we have not found other bacteria for testing. However, Sanger sequencing and conventional PCR were confirmed the current LAMP results as well.

4. In figure 2, authors showed their assay could detect 0.125 fg of DNA. Could authors dilute the template further to get to the point they could not see the bacterial DNA to know what would be the least amount of DNA required for its detection? Additionally, when diluting the target DNA could they also spike the samples with non-specific DNA to ensure The non-specific DNA does not mask H pylori detection.

-  Meanwhile the validation step, we have diluted the DNA to measure the LOD, and we have also spiked DNAs of  HPAG1 and DU-30 strains to healthy human DNA (non-specific DNA) to analysis the H.Pylori DNA detection as well. The results were approved and acceptable.

Round 2

Reviewer 1 Report

the major issue is that the authors insisted that the use of LAMP is enough as gold standard in H. pylori discussion. I do not agree with this because the authors are using a circular definition. The use of conventional PCR or Sanger is only to confirm the finds with LAMP. In order to assess the sensitivity and specificity of any assay the general approach is compare the results of a well validate diagnostic assay with the testing assay. If the authors do not have any other validated test to compare with they need to clarify it in the text.

In its current form the results are only relative to LAMP assay

Author Response

Dear Editors,

We would like to thank reviewers for suggestions.

The manuscript is re-revised regarding to the reviewers comments. The corrections are mentioned as follows and by yellow highlight style throughout the manuscript to better clarification.

We are looking forward to hearing from you

Yours Sincerely,

Reviewer Comment

the major issue is that the authors insisted that the use of LAMP is enough as gold standard in H. pylori discussion. I do not agree with this because the authors are using a circular definition. The use of conventional PCR or Sanger is only to confirm the finds with LAMP. In order to assess the sensitivity and specificity of any assay the general approach is compare the results of a well validate diagnostic assay with the testing assay. If the authors do not have any other validated test to compare with they need to clarify it in the text. In its current form the results are only relative to LAMP assay.

  • More explanations are added and mentioned into discussion section for better clarification.

The manuscript result suggested oral specimens (noninvasive sampling) might be considered for H.pylori detection as gold standard sampling instead of biopsy (invasive sampling).

We do not think the LAMP assay is the gold standard for H.pylori detection right now. Our finding supported that LAMP assay can be considered as a valid test (a point of care testing) for screening and detection of H.pylori in gastrointestinal dysfunctions in clinical settings especially in low-income countries.

In addition, results were confirmed by Sanger sequencing and conventional PCR to verify the current outcomes. However, it is recommended to compare the current results by a CE, IVD diagnostic kit for final verification of the results in future studies.

Round 3

Reviewer 1 Report

Thank you for your modifications and to my comments